# Interoperability as a Catalyst for Digital Health and Therapeutics: A Scoping Review of Emerging Technologies and Standards (2015–2025)

**DOI:** 10.3390/ijerph22101535

**Published:** 2025-10-08

**Authors:** Kola Adegoke, Abimbola Adegoke, Deborah Dawodu, Akorede Adekoya, Ayoola Bayowa, Temitope Kayode, Mallika Singh

**Affiliations:** 1School of Health Sciences and Practice, New York Medical College, 40 Sunshine Cottage Road, Valhalla, NY 10595, USA; tkayode@student.touro.edu (T.K.); msingh25@student.touro.edu (M.S.); 2Department of Health & Biomedical Sciences, College of Health Professions, University of Texas, Rio Grande Valley, One West University Blvd, Brownsville, TX 78520, USA; abimbola.adegoke@uth.tmc.edu; 3McWilliams School of Biomedical Informatics, UTHealth Houston, 7000 Fannin Street, Houston, TX 77030, USA; deborah.dawodu@uth.tmc.edu; 4Department of Global Health, McMaster University, 1280 Main St W, Hamilton, ON L8S 4L8, Canada; akoredeadekoya.f@gmail.com; 5Department of Business Administration, Thompson Rivers University, Kamloops, BC V2C 0C8, Canada; bayowaayoola@gmail.com

**Keywords:** digital health, interoperability, digital therapeutics, emerging technologies, blockchain, FHIR, LMICs, health data standards

## Abstract

Background: Interoperability is fundamental for advancing digital health and digital therapeutics, particularly with the integration of technologies such as artificial intelligence (AI), blockchain, and federated learning. Low- and middle-income countries (LMICs), where digital infrastructure remains fragmented, face specific challenges in implementing standardized and scalable systems. Methods: This scoping review was conducted using the Arksey and O’Malley framework, refined by Levac et al., and the Joanna Briggs Institute guidelines. Five databases (PubMed, Scopus, IEEE Xplore, ACM Digital Library, and Google Scholar) were searched for peer-reviewed English language studies published between 2015 and 2025. We identified 255 potentially eligible articles and selected a 10% random sample (n = 26) using Stata 18 by StataCorp LLC, College Station, TX, USA, for in-depth data charting and thematic synthesis. Results: The selected studies spanned over 15 countries and addressed priority technologies, including mobile health (mHealth), the use of Health Level Seven (HL7)’s Fast Healthcare Interoperability Resources (FHIR) for data exchange, and blockchain. Interoperability enablers include standards (e.g., HL7 FHIR), data governance frameworks, and policy interventions. Low- and Middle-Income Countries (LMICs) face common issues related to digital capacity shortages, legacy systems, and governance fragmentation. Five thematic areas were identified: (1) policy and governance; (2) standards-based integration; (3) infrastructure and platforms; (4) emerging technologies; and (5) LMIC implementation issues. Conclusions: Emerging digital health technologies increasingly rely on interoperability standards to scale their operation. Although global standards such as FHIR and the Trusted Exchange Framework and Common Agreement (TEFCA) are gaining momentum, LMICs require dedicated governance, infrastructure, and capacity investments to make equitable use feasible. Future initiatives can benefit from using science- and equity-informed frameworks.

## 1. Plain Language Summary

This review examines how digital health tools such as mobile apps, artificial intelligence, and blockchain can be integrated more effectively through standard rules and systems, known as interoperability. By studying research from 2015 to 2025, we found that using standards such as FHIR and HL7 helps digital tools “talk” to each other more effectively. However, in lower-income countries, limited Internet access, outdated systems, and a lack of policies further complicate this. We reviewed 26 studies and identified five major areas that require attention, including the following: the development of more effective policies, the establishment of robust digital infrastructure, and the provision of equitable access for all communities. This work can help health leaders and technology developers build interconnected systems to enhance patient care globally.

## 2. Introduction

### 2.1. Background

#### 2.1.1. Interoperability as a Foundational Enabler of Digital Health

Digital health technologies promise personalized data-driven care. However, fragmented information systems create significant barriers for effective communication. As Lehne and colleagues emphasize, when systems cannot seamlessly exchange data, valuable digital innovations (such as Artificial Intelligence (AI) and big-data analytics) remain “hidden in isolated databases” and fail to reach their potential [1]. This is especially acute in low- and middle-income countries (LMICs) where diverse donor-driven projects often lack a common set of standards. For instance, recent analyses note that in many LMIC settings, “fragmented systems, limited digital infrastructure, and the absence of standardized health records hinder progress” [2]. Without interoperability, redundant testing and administrative waste abound, ultimately undermining all efforts to create patient-centric, scalable, and safe healthcare ecosystems [1,2].

#### 2.1.2. Policy Push and International Frameworks

Global policy initiatives are moving toward greater emphasis on interoperability. The World Health Organization’s (WHO) Global Strategy on Digital Health (2020–2025) advocates for the use of open standards and the sharing of structured data to support improvements in the healthcare system [3]. The states’ ONC’s Trusted Exchange Framework and Common Agreement (TEFCA) established nationwide rules and technology standards for health information exchanges [4]. Other countries are building comparable national and regional data-exchange frameworks. Implementation has not yet caught up with this policy. As noted by the WHO, the attainment of interoperability varies greatly by context, depending on national strategy, governance arrangements, infrastructure, and other factors [5].

#### 2.1.3. Standards-Based Interoperability Infrastructure

APIs and standards are the blocks of most interoperability efforts. More specifically, the HL7 FHIR standard has gained global traction as it is “widely implemented by vendors and healthcare systems” and enforced by regulatory agencies worldwide [6]. The API-first philosophy employed by FHIR supports modular integration, and platforms such as SMART facilitate the native integration of third-party applications across multiple EHR platforms. These standards-based platforms are transforming health IT in a way that enables disparate vendor systems to communicate with each other, share patient records, and support coordinated care [7,8].

#### 2.1.4. Emerging Technologies and Interoperability

New technologies have been introduced into healthcare systems, each with its own implications for interoperability. Strong decision support is offered by machine learning and artificial intelligence; however, this comes with the resulting governance issues, which depend on the availability of high-quality interoperable data. Blockchain and distributed ledger technologies offer decentralization and data integrity, albeit with tradeoffs in terms of privacy and scalability. For example, blockchain systems may bottleneck large volumes of transactions such that strong patient privacy rules will have to be traded off against the transparency of shared ledgers [9]. However, Internet of Things (IoT) devices and federated learning frameworks provide rich data streams, necessitating the establishment of new standards and trust models. They promise good personalization and automation but introduce new layers of complexity to the architectural and regulatory aspects of interoperability [10,11].

#### 2.1.5. Interoperability in Digital Therapeutics and Chronic Care

Interoperability is crucial in areas such as digital therapeutics and chronic disease care, where care increasingly relies on applications, wearable devices, and remote monitoring. If integrated smoothly, patient-reported data (e.g., from sensors or smartphone apps) can influence clinician decisions and support ongoing care. The scalability of these digital therapeutic solutions remains elusive. Studies have noted how most DTx systems have been found to reside in silos that are disconnected from electronic health records and systematic care teams and are therefore not scalable [12]. Without integration and data standards, these tools are not scalable or capable of providing end-to-end care and thus cannot effectively impact outcomes in chronic diseases [13].

### 2.2. Rationale

Despite the high-priority status of interoperability, no integrated synthesis exists on how various technologies, standards, and governance models are deployed in digital health across diverse settings. Existing reviews tend to narrowly focus on one technology or geographical region, leaving crosscutting issues inadequately addressed. Ongoing challenges, including inadequate infrastructure, weak governance, and a limited workforce, delay the development of interoperable systems in most LMICs [2]. Information on global standard harmonization, such as FHIR, or new tools, such as blockchains, is lacking in these settings. Systematic mapping of the literature is crucial for identifying gaps and guiding future investments in interoperable digital health.

### 2.3. Objectives and Research Questions

This scoping review aims to map the breadth of the literature (2015–2025) on digital health interoperability, with a particular emphasis on emerging technologies, data standards, and governance models both globally and in LMICs. In particular, we address the following issues.

Emerging technologies: Which new technologies (e.g., AI, blockchain, IoT, federated learning) are being integrated to support digital health interoperability?Standards and Governance: What standards for data (e.g., HL7 FHIR, open APIs) and governance models (global, national, vendor-led) are implemented to enable interoperability?Enablers and Barriers: What are the most important enablers and inhibitors of interoperable HIS implementation, particularly in LMIC contexts?

These questions will be examined through a scoping review, following established methodologies. Contemporary ethical and regulatory standards have informed the latter. For example, the WHO report on AI Ethics insists that all systems of AI and data sharing “must put ethics and human rights at the core of [their] design, deployment, and use’ [14]. Through the integration of technological, policy, and implementation perspectives, our study reveals the current practices and imperatives for maximizing interoperability across global health.

## 3. Methods

### 3.1. Study Design

This scoping review was conducted in accordance with the instructions of Arksey and O’Malley [15], later revised by Levac et al. [16], and subsequently by the Joanna Briggs Institute (JBI) [17]. The PRISMA-ScR checklist [18] was used for reporting. The review was registered in the Open Science Framework (OSF) with the full protocol and data appendices (DOI: https://doi.org/10.17605/OSF.IO/YBAX3, Appendix A).

### 3.2. Objectives and Design

The objective was to map the global landscape of digital health interoperability frameworks by highlighting emerging technologies, governance frameworks, and equity considerations, particularly across the LMICs. Given the variation in approaches and technologies, a scoping review was selected owing to its suitability in compiling divergent conceptual and empirical literature across fields, such as digital health, policy, and informatics.

This review followed a multi-phase approach:Protocol development (Appendix A) and team calibration;A systematic search was conducted on five databases;Screening Title, abstracts, and full texts;Random selection of a 10% sample for detailed extraction;Thematic synthesis and statistical representativeness testing.

### 3.3. Eligibility

Eligibility was guided by the PCC (Population–Concept–Context) framework:Population: Health system users and digital governance entities;Concept: Interoperability frameworks, standards (e.g., HL7 and FHIR), and new technologies (e.g., blockchain and artificial intelligence);Context: Global studies with a focus on LMICs or inclusive settings.

Inclusion Criteria:Peer-reviewed journal or conference papers;Addressing technical or semantic interoperability in digital/uHealth;Implementation of at least one emerging technology (AI, blockchain, federated learning, and IoT);Discussion about governance, standards, or equity implications;January 2015 through June 2025 published writings;English language.

Exclusion Criteria:Preprints, grey literature, or non-peer-reviewed work;Single studies about radiology or imaging-related artificial intelligence;Articles limited to high-income countries without relevance to LMIC or global contexts;Concept papers without implementation or technical details.

### 3.4. Information Sources and Search Strategy

A careful search was conducted via six electronic databases:PubMed;Scopus;IEEE Xplore;ACM Digital Library;Google Scholar (hand-searched).

Searches were conducted in September 2025, focusing on publications published between 2015 and 2025.

Search terms included the following:“digital health” OR “uHealth” OR “mobile health” OR “digital therapeutics”“interoperability” OR “FHIR” OR “HL7” OR “TEFCA” OR “data governance”“blockchain” OR “AI” OR “artificial intelligence” OR “federated learning” OR “IoT”“LMIC” OR “low- and middle-income country” OR “global health”

Boolean logic is tailored to each database. See Appendix B for the full search strings. Records were exported to Zotero and EndNote and uploaded to Covidence^®^ (Veritas Health Innovation, Melbourne, Australia) for deduplication and screening.

### 3.5. Selection Process

Screening was a three-tiered process:Title and abstract screening was conducted by [K.A.] and [T.K.] individually;The full-text assessment was conducted on eligible or questionable records;Disagreements were resolved by consensus with [D.D., A.B.].

A total of 417 studies were identified, 255 of which met the inclusion criteria after full-text screening. This process is summarized in Appendix C (PRISMA flow diagram) [18] and Figure 1.

Inter-rater reliability was assessed for both the title/abstract and full-text screening. Agreement was high, with Cohen’s κ = 0.82 for title/abstract screening and κ = 0.79 for full-text screening, indicating substantial reliability between reviewers.

### 3.6. Sample Selection and Data Charting

Owing to the large number of eligible studies, feasibility and resource constraints dictated that detailed data extraction and thematic coding were performed only on a 10% random subsample (n = 26), which was then selected for in-depth analysis. Although initially planned for stratified purposeful sampling [19,20], a uniform randomization strategy was implemented using the Stata Command (Appendix F) and output (Appendix G). Randomization was validated to ensure regional and thematic representations.

Data were extracted using a structured matrix aligned with JBI guidance [17]. Variables included the following:Study title, author(s), and year;Country/region;Intervention and the digital health domain;Technology used (e.g., AI, blockchain, and IoT);Interoperability standard(s) (e.g., HL7 FHIR, and TEFCA);Governance or regulatory frameworks;Barriers and enablers;Mention of equity or inclusion;Additional observations.

Extraction was performed using [K.A.] and [T.K.], and independently validated using [D.D., A.A. (Abimbola Adegoke)] for consistency. See Appendix D for the full extraction matrix, and Appendix E for the complete dataset of the 26 studies.

### 3.7. Synthesis and Analysis

Synthesis was performed using inductive thematic analysis, guided by frameworks from Braun and Clarke (2006) and aligned with best practices in scoping reviews [16,17]. Key themes were developed and refined iteratively across the five domains.

Governance and Policy Frameworks;Interoperability Standards and Technical Architectures;Emerging Technologies;Barriers and Enablers to Implementation;Equity and LMIC Representation.

The findings are further illustrated with figures that enhance the data presented in the tables. Figure 2 illustrates the citation frequency of key interoperability technologies, with HL7 FHIR and AI/ML being the most frequently cited, followed by blockchain, SNOMED CT, and LOINC. Figure 3 illustrates the geographic distribution of studies across LMIC, HIC, and international settings, highlighting the balance of regional contributions. Figure 4 encapsulates the reported barriers and enablers to interoperability, ranging from legacy systems and semantic mismatches to open standards and co-designed governance frameworks. Individually, and collectively with Figure 2, Figure 3 and Figure 4, these visualizations afford a comparative technology mapping, regional patterns, and a set of implementation difficulties.

### 3.8. Statistical Justification and Representativeness

We assessed the representativeness of the 10% sample.

#### 3.8.1. Distributional Comparison

Full dataset: 255 studies (2006–2025); Sample: 26 (2019–2025).Journal articles: 92.3% in sample vs. 72.9% in full dataset.Conference papers: 7.7% in sample vs. 27.1% in full dataset.Noted overrepresentation of journal articles in the sample.

#### 3.8.2. Chi-Square Test

Conclusion: There was no significant difference between the sample and full set; the random sample was representative of the intervention categories.

### 3.9. Limitations

As with all scoping reviews, several methodological limitations are acknowledged:Subsampling: Only 10% of the eligible studies were included because of feasibility constraints. Although random sampling was statistically validated (χ^2^ = 1.25, *p* = 0.535), thematic saturation could not be guaranteed.Language Restriction: Only English-language publications were included, excluding potentially relevant studies in French, Spanish, or local LMIC.Source Restrictions: Gray literature and non-indexed implementation reports were excluded, which may have resulted in the underrepresentation of field-based and NGO-led initiatives.Database and Year Limits: Although the databases were diverse, some relevant publications indexed after the search (June 2025) may have been missing.

Nonetheless, these methods align with the best practice recommendations for breadth-focused evidence mapping [15,16,17].

## 4. Results

### 4.1. Selection of Sources of Evidence

A total of 417 records were identified through searches of PubMed, Scopus, IEEE Xplore, ACM Digital Library, and Google Scholar databases. After de-duplication using Zotero, EndNote, and Covidence, 255 studies were retained after full-text screening (Appendix D). In line with the scoping review guidance, we present descriptive statistics for the full set of eligible studies, including publication year, type, and domain focus (Table 1).

Owing to feasibility and resource constraints, detailed data extraction and thematic coding were performed only on a 10% random subsample (n = 26). This subsample serves as a supplementary analytic layer for the in-depth synthesis of interoperability enablers, barriers, standards, and equity considerations. The subsample was validated as broadly representative of the 255-study pool across year and publication type distributions. The reasons for exclusion at each stage (e.g., duplication, relevance mismatch, or lack of interoperability focus) are documented in the PRISMA-ScR flow diagram (Appendix C).

As shown in Table 1, 255 eligible studies spanned nearly two decades (2006–2025), with the majority published between 2022 and 2025. Journal articles (70.6%) predominated over conference papers (24.3%), with a small number remaining unclassified. In terms of domain focus, digital/mHealth tools (47.8%), artificial intelligence/machine learning (24.7%), and blockchain (22.0%) were the most common, whereas relatively few addressed standards, IoT, or digital therapeutics.

While these descriptive characteristics are presented for the entire 255-study pool, detailed thematic extraction was conducted only on the 10% subsample (n = 26), which served as a supplementary analytic layer for in-depth synthesis [16,17].

The table summarizes the bibliographic characteristics of the 255 studies that met full-text eligibility criteria, including publication year, type of publication, and domain focus. Detailed thematic extraction was not conducted for these studies beyond bibliographic coding; a 10% subsample (n = 26) was used for in-depth synthesis (see Section 4.5).

### 4.2. Characteristics of Included Studies

The final 26 studies included in the synthesis spanned the global (n = 7), LMIC (n = 10), and HIC (n = 9) contexts. Study types included the following:Scoping/Systematic Reviews: 9;Conceptual or Framework Papers: 3;Prototype/Pilot Studies: 3;Viewpoint or Qualitative Analyses: 11.

The technological domains included AI/ML (n = 16), blockchain (n = 9), FHIR/standards (n = 13), semantic ontology frameworks (Systematized Nomenclature of Medicine—Clinical Terms) (SNOMED CT), Observational Medical Outcomes Partnership Common Data Model (OMOP CDM), and Logical Observation Identifiers Names and Codes (LOINC).

Charted data included the following:Title, authorship, year, and region;Technology focus and interoperability standard;Barriers, enablers, and equity considerations;Governance/policy and implementation themes.

Each study is cited in Section 4.1, Section 4.2, Section 4.3, Section 4.4, Section 4.5 and Section 4.6 and detailed in the extraction matrix (Appendix E).

### 4.3. Critical Appraisal Within Sources of Evidence

As this was a scoping review, no formal critical appraisal (e.g., risk of bias assessment) was performed. However, the inclusion of only peer-reviewed studies ensured a minimum quality threshold. The methodological type and transparency were considered during the thematic synthesis process, and studies lacking technical or contextual clarity were excluded at the eligibility screening stage (Section 3.3).

### 4.4. Results of Individual Sources of Evidence

Each of the 26 studies is discussed across the five main thematic domains in Section 4.1, Section 4.2, Section 4.3, Section 4.4 and Section 4.5. The table below summarizes the results across studies, with direct citations aligned with the synthesis, as presented in Table 2 and Table 3 (see the extraction matrix in Appendix E for full details).

The results are presented in both narrative and tabular formats, supported by Table 1, Table 4 and Table 5. Table 1 summarizes the descriptive characteristics of all 255 eligible studies, while Table 4 and Table 5 detail the thematic coverage and study-level synthesis of the 26 studies that underwent detailed extraction.

Summary Patterns and Trends

Across 26 studies, the following were attained:AI/ML appeared in 16;Blockchain in 9;HL7 FHIR in 13;Equity considerations in 9;Barriers/enablers in 13.

Total Counts from the Table:Governance (4.1) → 6 studies;Standards (4.2) → 8 studies;Emerging Tech (4.3) → 14 studies;Barriers/Enablers (4.4) → 13 studies;Equity (4.5) → 9 studies.

Each study appears in at least one result subsection. Some appeared in multiple categories to reflect their relevance across various areas.

### 4.5. Synthesis of Results

#### 4.5.1. Governance and Standards

Six studies highlighted the use of digital governance frameworks to facilitate interoperability, particularly in LMICs, where policy fragmentation and weak regulatory capacity hinder development [21,24]. It finds an echo in high-income settings, where fragmented reimbursability and regulatory practices persist; the most organized model available is the German DiGA route, but it has found patchy replication in the rest of Europe [31]. From a system design perspective, enterprise architecture frameworks have been developed to align clinical and IT systems using structured governance, thereby enhancing patient safety [22].

National electronic plans, such as Botswana’s eHealth strategy and broader interoperability plans (including Confluent HL7 FHIR, SNOMED CT, and OpenHIE), provide promising templates for electronic reform. Nevertheless, entrenched challenges, such as semantic mismatches and limited cross-system data sharing, as well as usability barriers, continue to hinder execution and scaling within applications [23,25].

Implementation of interoperability standards is often hindered by semantic mismatches, customization burdens, and incompatibility with legacy systems [32,33,34,35], although widely adopted frameworks such as HL7 FHIR (*n* = 13), along with SNOMED CT, OMOP CDM, and ISO 18308 [34,35,36], are used across diverse applications from sepsis detection to cancer data harmonization.

#### 4.5.2. Emerging Technologies and Innovation

Emerging technologies played a prominent role in the 14 studies examined. Blockchain was highlighted as the central theme in several studies [26,27,30,37], offering enhancements in data integrity, access control, and auditability—particularly in environments of low trust. Integrations of HL7 FHIR and Ethereum protocol-based protocols were suggested to facilitate informed data exchange in LMICs [26,37]. AI and machine learning were utilized for disease modeling, remote monitoring, and cybersecurity alerting [29,38,39,40], although limitations related to bias, data set representativeness, and language exclusion were reported [29,40]. Other innovations included emotional AI, health monitoring using the Internet of Things, as well as robotics for delivering tailor-made care [41,42]. A permissioned version of the blockchain, utilizing open health data, was also suggested for facilitating increased openness and secure data sharing [43].

#### 4.5.3. Implementation Barriers and Enablers

Thirteen studies identified recurring implementation barriers:Legacy infrastructure and system heterogeneity [32,35];Lack of workforce capacity and digital literacy, especially in LMICs [24,25,44];Legal fragmentation and policy misalignment [21,22,23];Semantic/API mismatches across platforms [33,34].

Conversely, enablers included the following:Open standards (FHIR, OpenHIE) [25,36];Co-designed governance and regional collaboration [23,24,45,46];NGO/donor support for local adaptation [28];Enterprise architecture models in institutional settings [31].

#### 4.5.4. Equity and Inclusion

Only 9 studies explicitly addressed equity. LMIC-focused analyses have highlighted that digital health systems often lack equity frameworks, particularly in donor-funded programs [28]. Risks included the following:Bias in AI algorithms due to non-diverse training data [40];Exclusion of rural/low-literacy populations [24,44];Language and access barriers in oncology and decision-support tools [29].

The proposed solutions include patient-controlled identity models [46], inclusive ontologies [35], and region-specific capacity building methods [24].

### 4.6. Summary Statistics and Patterns

Regions: 10 LMIC, 9 HIC, 7 Global.The most common standards are HL7 FHIR [13], SNOMED CT [6], and LOINC [3].Technology prevalence: AI/ML [16], Blockchain [9].Equity mentioned: 9 studies.The most frequent themes were implementation barriers (n = 13), governance gaps (n = 6), and integration of emerging technologies (n = 14).

A chi-square analysis comparing the LMIC and HIC studies showed no significant differences in theme prevalence (χ^2^ = 1.25, df = 2, *p* = 0.535), indicating similar interoperability challenges across income settings.

Legend Summary:

Figure 2, Figure 3 and Figure 4 collectively show that interoperability in digital health hinges on a few core technologies (FHIR, AI/ML), faces recurring barriers (legacy systems and policy gaps), and is supported by open standards, governance frameworks, and inclusive design. LMICs and HICs report remarkably similar patterns, although they tend to emphasize donor engagement and mobile solutions.

## 5. Discussion

### 5.1. Summary of Key Findings

This scoping review identified 255 studies with a conceptually diverse sample of 10% (n = 26) that were analyzed in depth. These findings indicate a proliferation of technical experimentation, and implementation maturity remains low, particularly in LMICs. Although the HL7 FHIR is the most widely adopted interoperability standard, its implementation varies substantially by region and governance context [6,32,33].

Although blockchains and AI have been widely discussed, their real-world deployment remains limited. For instance, although Elangovan et al. [37] and Gupta et al. [26] have described theoretically robust blockchain architectures, only a few have been fully implemented or integrated into national systems. Equity and patient ownership were underrepresented, as only nine of the 26 studies meaningfully addressed underserved populations or inclusivity, despite their clear relevance in LMICs [22,28,44].

### 5.2. Contribution to the Literature

To our knowledge, this is the first scoping review to synthesize interoperability, global standards (e.g., FHIR and SNOMED), and emerging technologies such as blockchain, federated learning, and AI across both HIC and LMIC settings [6,10,27,33,35]. Previous reviews typically focused on either a single technology (e.g., blockchain in EHRs [27]) or a specific regional setting (e.g., sub-Saharan policy gaps) [21].

This study also applies to a structured equity lens, highlighting the disconnection between digital health design and marginalized populations, particularly in AI-enabled systems where language and cultural assumptions may embed bias [29]. By evaluating both the technical and governance frameworks, this review offers new insights into implementation maturity, a dimension often overlooked in the conceptual literature.

### 5.3. Policy & Practice Implications

Findings have immediate relevance for both global health policy and digital health system design:LMICs require contextualized standards; for example, Botswana’s adoption of HL7 FHIR and SNOMED demonstrates that national standardization is feasible even in resource-constrained settings but necessitates governance alignment and capacity building [25].Cross-border frameworks must evolve, as projects in Europe [22], Latin America [24], and sub-Saharan Africa [21] demonstrate that legal, semantic, and technical fragmentation hinder the scalability and security of data sharing.Developers must design with integration and scalability in mind: As highlighted in Wong et al.’s enterprise architecture framework [31], design decisions will need to factor in regulatory realities, legacy systems, and multilingual environments to avoid siloed implementations.

Furthermore, developers and policymakers would need to integrate equity principles from the outset rather than view them as downstream afterthoughts [14,28].

### 5.4. Future Research Agenda

The review identifies several key areas for future research:Longitudinal evaluation of blockchain-FHIR hybrids: Several prototypes have been detailed in research studies, although relatively few have tracked outcomes or patient impact longitudinally [30,34,37].Implementation science for LMIC settings: There is a clear need for empirical studies using mixed methods to understand what enables or prevents the uptake of interoperability in resource-poor settings [23,44].Maturity models for interoperability: Currently, few frameworks exist to quantify the level of interoperability development across health systems [24,31]. The standard model facilitates benchmarking.Move beyond conceptual frameworks: Published articles include theoretical models and prototypes from preliminary stages [37,39,41]. Further studies are required to address deployment in real-world settings, scalability, and cost-effectiveness.

### 5.5. Limitations

The evidence found by this review needs to be considered alongside these particular limitations:Although a 10% sample was statistically representative, it would not necessarily fully reflect the regional or thematic richness, particularly for novel or emerging technologies, such as federated learning, or less common standards, such as FHIR.Omitting gray literature could underestimate ground-level implementation efforts, particularly in LMIC settings where documentation is frequently located outside academic publishing.The English-only inclusion criteria may have excluded studies that recorded context-related interoperability efforts in Francophone Africa, Latin America, or Southeast Asia.The bulk of the published work is conceptual or prototype-based in nature; therefore, it limits inferences regarding real-world scalability and long-term consequences.

Despite these limitations, this review provides a novel synthesis of interoperability across technologies, governance models, and settings and helps identify key gaps for future empirical work.

Nevertheless, these limitations are common in scoping reviews and are mitigated through rigorous searches, dual-reviewer screening, and equity-conscious synthesis [16,17,20].

## 6. Conclusions

Interoperability remains a fundamental facilitator of scalable, patient-centered, equitable health and digital health environments. Although novel advances are innumerable in new technologies such as HL7 FHIR, blockchain, and artificial intelligence, their practical implementation remains profoundly fragmented and inconsistent, particularly in middle- and low-income countries. This review highlights the disconnect between technological capabilities and real-world deployments. Fragmented governance, a lack of standardization, and limited consideration of equity and context are common barriers to effective governance. Conversely, open standards, strong policy regimes, and co-designs among parties signal more harmonized and equitable systems. To unlock the full promise of digital health innovation, the sector must move beyond experimentation and invest in shared governance, interoperability maturity models, and context-aware implementation. Future efforts should focus on equity, scalability, and sustainability to fulfill the promise of digital health systems.

## Figures and Tables

**Figure 1 ijerph-22-01535-f001:**
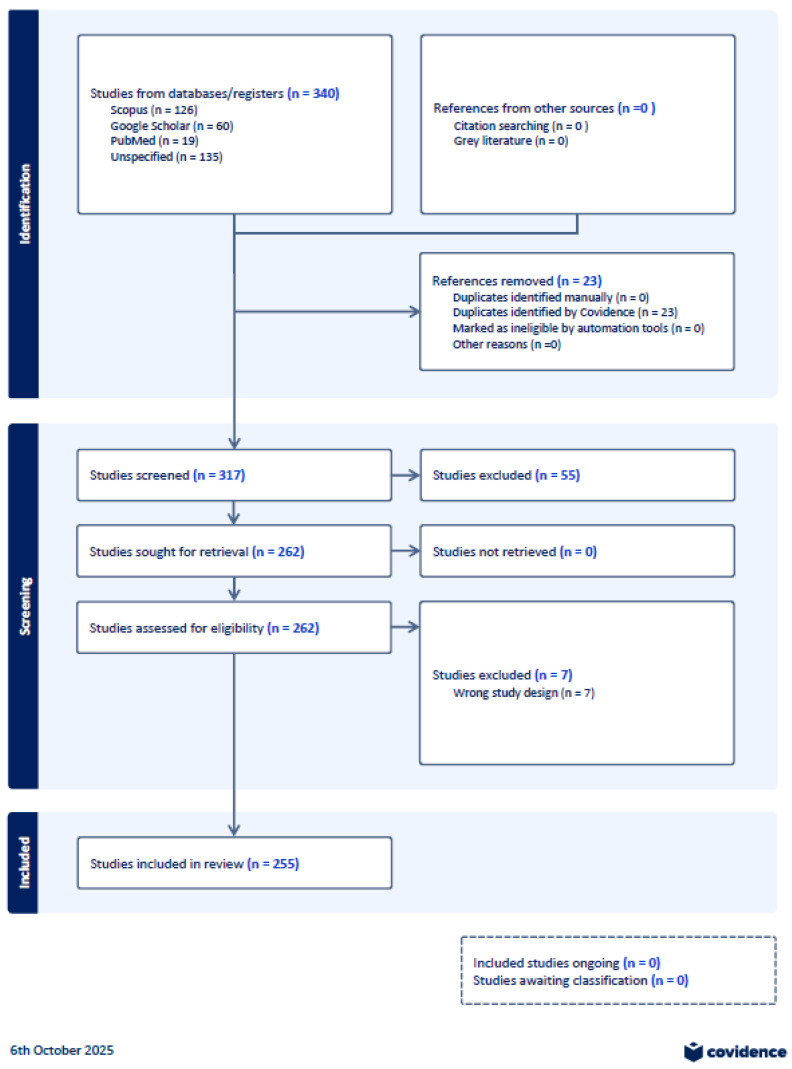
From the 255 eligible studies, a stratified random sample of 10% (n = 26) was selected using Stata’s sample function, ensuring thematic and regional diversity. This sample was used for detailed data charting and thematic synthesis.

**Figure 2 ijerph-22-01535-f002:**
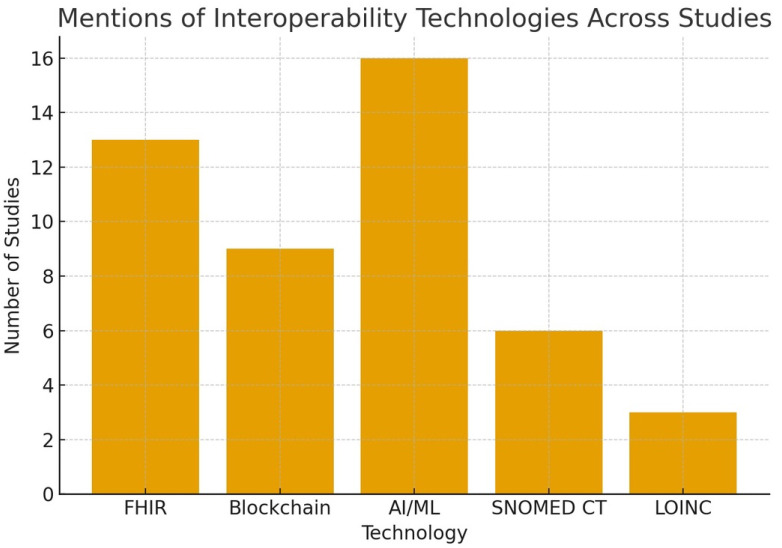
Frequency of studies citing interoperability technologies. HL7 FHIR (n = 13) and AI/ML (n = 16) were most frequently referenced, followed by blockchain (n = 9), SNOMED CT (n = 6), and LOINC (n = 3).

**Figure 3 ijerph-22-01535-f003:**
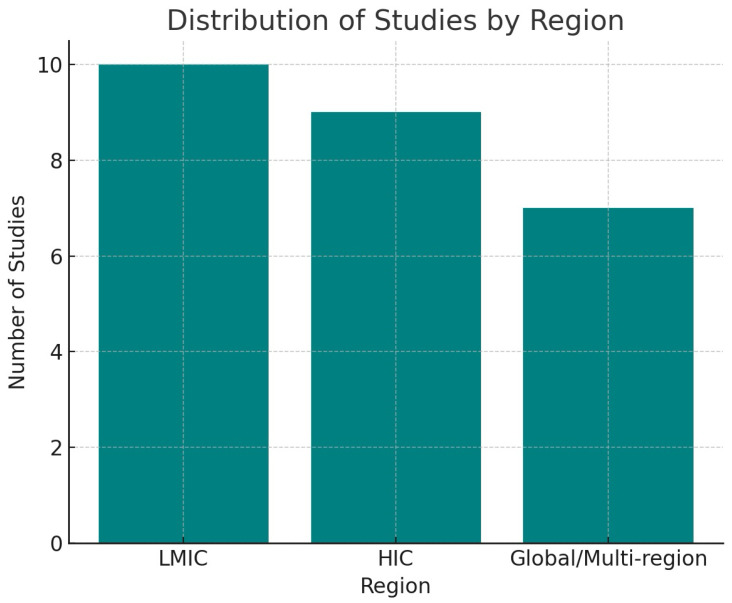
Distribution of studies by region. Ten studies focused on LMICs, nine on HICs, and seven had a global or multi-regional scope.

**Figure 4 ijerph-22-01535-f004:**
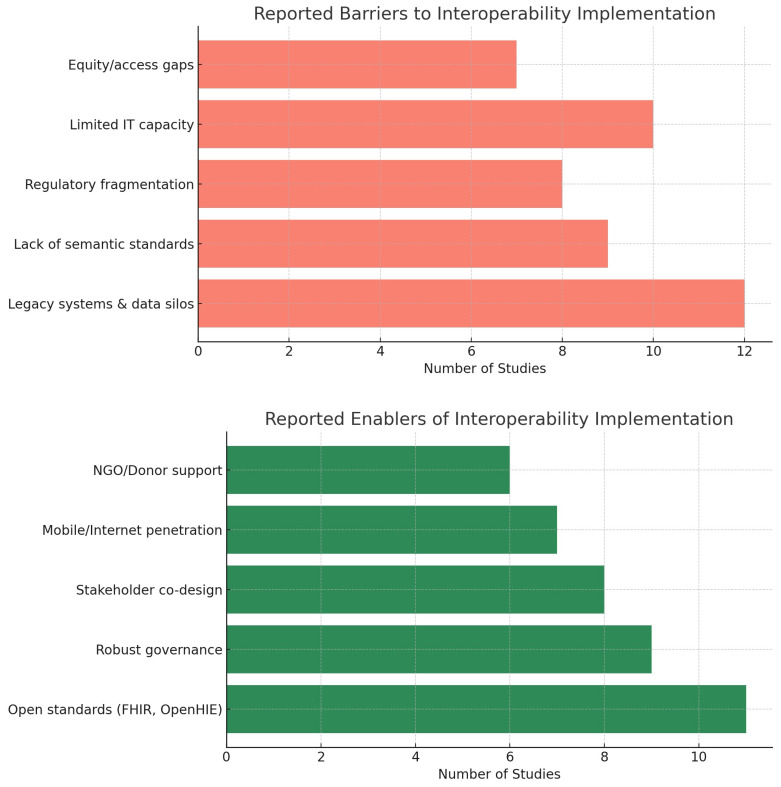
Reported barriers and enablers for interoperability. Barriers included legacy systems (n = 12), limited IT capacity (n = 10), semantic mismatches (n = 9), regulatory fragmentation (n = 8), and digital divides (n = 7). Enablers included open standards like FHIR/OpenHIE (n = 11), strong governance (n = 9), stakeholder co-design (n =8), infrastructure availability (n = 7), and NGO/donor support (n = 6).

**Table 1 ijerph-22-01535-t001:** Descriptive characteristics of all eligible studies (n = 255).

Characteristic	n (%)
Publication Year	
2006–2010	1 (0.4)
2011–2015	2 (0.8)
2016–2020	30 (11.8)
2021	14 (5.5)
2022	34 (13.3)
2023	29 (11.4)
2024	60 (23.5)
2025	85 (33.3)
Publication Type	
Journal Article	180 (70.6)
Conference Paper	62 (24.3)
Unknown/Not classified	13 (5.1)
Domain Focus	
Digital/mHealth/Other	122 (47.8)
Artificial Intelligence/ML	63 (24.7)
Blockchain	56 (22.0)
FHIR/Standards	7 (2.7)
Internet of Things (IoT)	6 (2.4)
Digital Therapeutics	1 (0.4)

Note: Descriptive statistics were based on the bibliographic characteristics of all 255 eligible studies. Detailed thematic extraction was conducted only on the 10% subsample (n = 26), which is analyzed in subsequent sections.

**Table 2 ijerph-22-01535-t002:** A chi-square test of independence was used to compare the intervention types across the full.

Metric	Value
χ^2^	1.25
df	2
*p*	0.535

Note: A chi-squared test of independence was performed to assess whether the distribution of intervention types in the random sample (n = 26) differed significantly from that in the complete set (N = 255). The results were not statistically significant (*p* > 0.05), suggesting that the sample was representative of the intervention categories.

**Table 3 ijerph-22-01535-t003:** Distribution of Intervention Types in Sample (n = 26).

Category	n (%)
Digital/mHealth Tools	13 (50.0%)
Policy/Structural	8 (30.8%)
Community/Education	5 (19.2%)

Note: Intervention categories were thematically coded in accordance with the study’s focus. “Digital/mHealth Tools” encompasses mobile health platforms, AI, blockchain, and interoperability technologies; “Policy/Structural” encompasses governance models and national strategies; and “Community/Education” encompasses digital literacy, training, and user-centered approaches.

**Table 4 ijerph-22-01535-t004:** Thematic Coverage Across Included Studies.

Domain	No. of Studies
Governance & Policy (4.1)	6
Interop Standards (4.2)	8
Emerging Tech (4.3)	14
Implementation (4.4)	13
Equity (4.5)	9

Note: Themes are based on coded domains identified in the synthesis (Section 4.1, Section 4.2, Section 4.3, Section 4.4 and Section 4.5). Studies could cover more than one theme; therefore, the totals exceeded the number of articles included.

**Table 5 ijerph-22-01535-t005:** Summary of Selected Studies by Theme, Region, and Equity Inclusion.

Study	Region	Themes	Barriers	Enablers	Equity Mention
Bene (2024) [21]	SSA	4.1, 4.4	Policy gaps	—	No
Fassbender (2024) [22]	EU	4.1, 4.4	Legal fragmentation	GDPR	No
Ambalavanan (2025) [23]	Global	4.1, 4.3, 4.4	No standards	Co-design	No
Curioso (2020) [24]	LATAM	4.1, 4.3, 4.4, 4.5	Capacity gaps	Regional collab	Yes
Ndlovu (2021) [25]	Botswana	4.1, 4.2, 4.4, 4.5	Legacy tech	HL7/FHIR/OpenHIE	Yes
Gupta (2019) [26]	Asia	4.2, 4.3	Data integrity	Smart contracts	No
Hasselgren (2020) [27]	Global	4.2, 4.3	Risk	Blockchain + FHIR	No
Sylla (2022) [28]	LMIC	4.4, 4.5	Donor dependence	Local adaptation	Yes
Yung (2025) [29]	Global	4.3, 4.5	Language bias	AI localization	Yes
Mamun (2022) [30]	NZ	4.2, 4.3	EHR heterogeneity	HL7 v2/3, ISO	No

Note: Themes correspond to the domains defined in thematic synthesis. “Equity Mention” indicates whether the study explicitly addressed equity considerations. Refer to Appendix E for the complete table, which includes all 26 studies.

## Data Availability

Research data supporting this publication are available from the OSF repository located at https://doi.org/10.17605/OSF.IO/YBAX3.

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
