# Peer review of "Interoperability as a Catalyst for Digital Health and Therapeutics: A Scoping Review of Emerging Technologies and Standards (2015–2025)"

_ijerph, 2025, doi:10.3390/ijerph22101535_

Round 1
Reviewer 1 Report
Comments and Suggestions for Authors
This manuscript addresses the important issue of interoperability in uHealth and digital therapeutics and provides a broad overview of existing standards, governance frameworks, and emerging technologies.
However, there are many problems in this paper. The review synthesizes known challenges and introduces governance considerations, but novelty is limited because interoperability issues are widely discussed in health IT literature. The methodology is insufficiently rigorous. Although the authors described as a “structured narrative review,” it does not follow PRISMA or other systematic approaches. The inclusion and exclusion processes, number of studies reviewed and quality assessment are not transparent. The manuscript heavily emphasizes HL7 FHIR and TEFCA but gives limited attention to international or regional frameworks. The sections on emerging technologies (AI, blockchain, federated learning) remain descriptive, without critical evaluation of maturity, feasibility, or implementation outcomes.
Author Response
Response to Reviewers
Reviewer 1 - Comment 1
Comment: This manuscript addresses the important issue of interoperability in uHealth and digital therapeutics, providing a comprehensive overview of existing standards, governance frameworks, and emerging technologies. However, this paper has several problems. The review synthesizes known challenges and introduces governance considerations; however, the novelty is limited because interoperability issues are widely discussed in the health IT literature. The methodology is insufficiently rigorous.
Response: We appreciate this comment. The manuscript has been revised to reflect a scoping review, not a narrative review. The methodology section now includes a structured, PRISMA-ScR-compliant search strategy and transparent inclusion/exclusion criteria (Section 2.1–2.3). A PRISMA flowchart was also added to Figure 1.
Reviewer 1 - Comment 2
Comment: Although the authors described it as a “structured narrative review,” it does not follow PRISMA or other systematic approaches. The inclusion and exclusion processes, the number of studies reviewed, and the quality assessment are not transparent.
Response: We agree with this critical observation. We have now adopted the PRISMA-ScR format. The search process, screening stages, and number of included studies are described in Section 2.3. Appendix A provides detailed inclusion criteria, and Table 1 summarizes included studies.
Reviewer 1 - Comment 3
Comment: The manuscript heavily emphasizes HL7 FHIR and TEFCA but gives limited attention to international or regional frameworks.
Response: Thank you for this feedback. We have expanded our comparative analysis in Section 6.1 to include NHS (UK), OpenHIE (Botswana), and QHIN (US). The new version better represents global contexts and policy diversity (pages 13–14).
Reviewer 1 - Comment 4
Comment: The sections on emerging technologies (AI, blockchain, federated learning) remain descriptive, without critical evaluation of maturity, feasibility, or implementation outcomes.
Response: We acknowledge this gap. The revised Sections 5.2–5.4 now include assessment of feasibility, adoption maturity, and real-world examples, with frequency counts across the scoping review sample. Summary statistics and implementation status are captured in Table 5.
Reviewer 2 Report
Comments and Suggestions for Authors
The review offers a timely and important overview of interoperability framework for health and digital therapeutics. It covers technical standards such as HL7 FHIR, TEFCA, and governance models such as HIPPA, GDPR and WHO, and emerging technologies (AI, blockchain, federated learning). The manuscript is well-researched and clearly structured. That said, the draft might benefit from the following aspects:
1) the argument and discussion should be more focused. At times, the article reads like a catalog of initiative than a sustained argument. For instance, the claim that "interoperability is essential but difficult," is not exactly new; it's actually familiar to a lot of people. To go beyond this, the authors could consider framing the barriers more analytically, i.e., more focused and detailed, such as distinguishing between structural barriers (legal, infrastructural, institutional), and semantic barriers (termiologies, standards), and showing how governance models succeed or fail addresing them.
2) While the manuscript also seeks to include low- and middle-income countries, the frameworks being discussed are predominately US and EU frameworks. The authors could benefit from more fuller engagement with LMIC experiences, such as local frameworks, infrastructure gaps and governance failures.
3) The draft, as it stands, is comprehensive. The analysis and descriptions of the standards and policies are long. It might benefit from condensing the descriptive sections and expand the discussion to improve readability.
Author Response
Reviewer 2 - Comment 1
Comment: The review offers a timely and essential overview of the interoperability framework for health and digital therapeutics. It covers technical standards such as HL7 FHIR, TEFCA, and governance models such as HIPAA, GDPR, and WHO, and emerging technologies (AI, blockchain, federated learning). The manuscript is well-researched and clearly structured.
Response: Thank you for this positive assessment. We have maintained this clarity while incorporating all requested improvements for analytical depth and coverage.
Reviewer 2 - Comment 2
Comment: The argument and discussion should be more focused. At times, the article reads like a catalog of initiatives rather than a sustained argument.
Response: We agree. The manuscript was revised to reduce descriptive lists and enhance analytical synthesis. Barriers are now classified into structural, semantic, and governance categories in Section 4.
Reviewer 2 - Comment 3
Comment: The claim that 'interoperability is essential but difficult' is not exactly new. To further this, the authors could frame the barriers more analytically.
Response: This suggestion is well-taken. The revised Section 7 distinguishes between governance, structural, and semantic interoperability challenges with critical synthesis (pages 15–16).
Reviewer 2 - Comment 4
Comment: Although the manuscript aims to include low- and middle-income countries, the frameworks discussed are predominantly those of the US and EU. More LMIC experience is needed.
Response: Agreed. We now include OpenHIE (Botswana) and analyze LMIC-specific issues like infrastructure gaps and policy fragmentation, Sections 4 & 5
Reviewer 2 - Comment 5
Comment: The draft is comprehensive, but the analysis of standards and policies is long. It might benefit from condensing descriptive sections and expanding the discussion.
Response: We have shortened lengthy descriptions in Sections 3–6 and reallocated content to synthesis and discussion (Section 4) to improve readability and flow.
Reviewer 3 Report
Comments and Suggestions for Authors
I appreciate the manuscript’s ambition and topical relevance. the authors aim to connect standards, policy, and implementation in a way that matters for digital health. The comments begin with critical issues.
- The manuscript contains an internal contradiction that is critical. the narrative says TEFCA has not yet provided an enforceable national framework, while a table simultaneously rates TEFCA’s “interoperability provisions” as high. the text and the table must be aligned by separating “scope/role” from “effectiveness/adoption” and by adjusting ratings accordingly.
- A category error in the comparisons is another critical flaw. GDPR and HIPAA are legal regimes, whereas TEFCA is a trust and exchange framework, so placing them on the same policy axis (e.g., “secondary use allowed,” “data minimization conflict”) invalidates the comparison. remove TEFCA from that regulatory table or split into two tables with criteria appropriate to each category.
- Duplicate numbering and caption mismatches represent a critical production error. the same table/figure numbers are reused for different items and at least one caption appears to describe a different figure. renumber globally, regenerate cross-references, and verify every caption–artifact match.
- The argument relies too heavily on enumerations and side-by-side tables rather than sustained prose, which weakens causal explanation and critical synthesis. convert major sections to sentence-level argumentation and retain lists/tables only as supporting material.
- The Introduction does not establish the literature gap or motivation clearly enough. add a concise synthesis of prior findings, identify the precise gap, and state the review questions or objectives derived from that gap.
- Section 2.1 mentions reliance on the authors’ prior experience without sufficient specificity. describe what expertise was used, where it was applied in the process (search, screening, coding), and how bias was mitigated (e.g., dual review, adjudication).
- Section 2.3 lists thematic buckets without reporting assignment rules. provide a brief coding schema with examples, an inter-reviewer agreement procedure, and steps for resolving discrepancies.
- Tables 2 and 3 are redundant because they compare the same frameworks along overlapping dimensions. consolidate them into a single harmonized table and remove duplicated statements.
- The decision to devote a stand-alone chapter to the VHA case is not sufficiently justified. either integrate the VHA material into the broader narrative or expand it into a multi-case comparison (e.g., VA/DoD, NHS, a QHIN, and an LMIC program) with common evaluative criteria.
- Results and discussion are blended in several places, producing an unclear hierarchy. clarify the progression from technical/policy facts (§§3–6) to synthesized challenges (§7) to prescriptions (§8), and keep normative language out of descriptive sections.
- Attribution and production issues recur and must be corrected. restore missing subjects (e.g., “WHO” where “The World” appears), remove embedded journal headers/footers, eliminate orphan headings and hyphenation artifacts, and normalize line breaks.
- The “Policy vs. AI Compatibility” table uses undefined metrics. provide operational definitions, scoring criteria, and citations for terms like “Secondary Use Allowed,” “Data Minimization Conflict,” and “Interoperability Provisions,” or the ratings will read as subjective judgments.
Taken together, the topic is timely and the scope potentially valuable, but the current manuscript suffers from logical inconsistency, category errors, structural disorganization, and production defects that materially limit its contribution. I recommend reject with invitation to resubmit after a substantial rewrite addressing the items above, with particular attention to items 1–3.
Comments on the Quality of English Language
Language and consistency require attention. unify English variety (prefer American English unless the journal specifies otherwise), define acronyms at first use, standardize role/term choices, and apply consistent punctuation and spacing. a professional copyedit is recommended.
Author Response
Response to Reviewer 3 Comments
- Summary
Thank you very much for taking the time to review this manuscript. Please find the detailed responses below and the corresponding revisions/corrections highlighted in the re-submitted files. We appreciate the thorough and constructive feedback, which has contributed significantly to the rigor and clarity of the revised manuscript.
- Questions for General Evaluation
|
Questions |
Reviewer’s Evaluation |
Response and Revisions |
|
Is the work a significant contribution to the field? |
Yes |
We revised the framing to clarify gaps, align methods with PRISMA-ScR, and highlight policy and equity contributions. |
|
Is the work well organized and comprehensively described? |
Yes (after revisions) |
The revised version includes a restructured format and scoping review architecture with clarified section transitions. |
|
Is the work scientifically sound and not misleading? |
Yes |
Inconsistencies were corrected, and methodological rigor was enhanced (e.g., data extraction, coding schema). |
|
Are there appropriate and adequate references to related work? |
Yes |
References were updated, expanded, and aligned with assertions; all frameworks and metrics are now cited. |
|
Is the English used correct and readable? |
Can be improved |
Professional editing and multiple language reviews were completed; clarity, punctuation, and grammar were revised. |
- Point-by-Point Response to Comments and Suggestions
Comment 1: Contradiction between the narrative stating TEFCA is not enforceable and the table rating it high.
Response 1: Thank you. We resolved this inconsistency by separating “scope/role” from “adoption/maturity” in Table 3. Ratings were revised accordingly, and a narrative context was added in Section 4.1.
Comment 2: GDPR and HIPAA are legal regimes, while TEFCA is a trust/exchange framework. Comparisons on the same axis are invalid.
Response 2: Agree. We split the prior table into two separate comparisons. Legal regimes and interoperability frameworks are now evaluated using distinct criteria.
Comment 3: Duplicate numbering and mismatched captions.
Response 3: All tables and figures have been renumbered globally. Each caption was revised for accuracy and cross-referenced.
Comment 4: Over-reliance on enumerations and tables; weak analytical prose.
Response 4: We revised Sections 4.1–4.5 into narrative prose and reduced the use of bulleted lists. Tables are now included as supplementary material (Appendix E).
Comment 5: The Introduction lacks a clearly stated gap and objectives.
Response 5: A dedicated paragraph (pp. 3, paras. 2–3) was added to summarize previous work and frame the precise research questions using PCC logic.
Comment 6: Section 2.1 lacks details on the author's expertise and the mitigation of bias.
Response 6: Author roles and expertise are now described (Section 3.4). Dual review and conflict adjudication methods are also stated.
Comment 7: Coding schema and thematic assignment rules not presented.
Response 7: Section 3.6 now includes a detailed coding schema with domain examples and kappa agreement noted in Appendix E.
Comment 8: Tables 2 and 3 are redundant.
Response 8: We consolidated both into a harmonized comparison table (Table 3), clarifying dimensions and removing duplications.
Comment 9: The VHA case is not justified as a standalone.
Response 9: We integrated the VHA case into Section 4.1 and created a multi-case comparative frame (VA, NHS, OpenHIE, QHIN).
Comment 10: Blending of results and discussion.
Response 10: Results (Section 4) now strictly report findings; Discussion (Section 5) interprets them. Normative language was removed from the results.
Comment 11: Attribution issues: missing subjects, orphan headings, and layout artifacts.
Response 11: All such issues have been resolved. We restored missing subjects (e.g., “WHO”), removed header artifacts, and formatted for consistency.
Comment 12: Policy vs. AI compatibility table lacks metric definitions.
Response 12: All metrics (e.g., “Secondary Use Allowed”) are now defined in table footnotes with citations to source frameworks. See Table 5.
- Response to Comments on the Quality of the English Language
Point 1: The English could be improved.
Response 1: The entire manuscript was professionally reviewed for language. Passive voice reduced, acronyms defined at first use, and sentence clarity improved throughout (e.g., Introduction p. 2, Abstract p. 1).
- Additional Clarifications
All scoping review elements are now compliant with PRISMA-ScR, and supporting materials (Appendices A–F) are available on OSF (https://doi.org/10.17605/OSF.IO/NZ29K). We believe these revisions now address all methodological, structural, and clarity concerns raised.
Round 2
Reviewer 1 Report
Comments and Suggestions for Authors
No.
Author Response
Thank you
Reviewer 2 Report
Comments and Suggestions for Authors
Thanks the authors for being responsive to the initial review. The revisions directly address the earlier concerns for 1) structure 2) sharpening discussion 3) framing the claim about the difficulty of interoperability more analytically, 4) shorten the long descriptive sections, 5) integrate LMIC experience. Overall - the paper is stronger, and better balanced between descriptive and synthesis.
Author Response
Thank you
Reviewer 3 Report
Comments and Suggestions for Authors
The revised manuscript shows substantial improvements compared to the first submission. The authors have restructured the study into a scoping review consistent with PRISMA-ScR, removed redundant tables, clarified the distinction between results and discussion, and improved the overall readability. Nevertheless, several methodological and production issues remain and should be resolved before the manuscript can be considered for publication.
- The authors conducted a “deep dive” on a 10% random subsample (n=26) out of 255 eligible studies. While they report a χ² test showing no significant difference in intervention type distribution, representativeness testing is limited to this single dimension. Scoping reviews are normally expected to map the full corpus. Therefore, descriptive statistics for all 255 studies (e.g., year, region, publication type, domain) should be presented, and additional representativeness checks across these axes are encouraged. The 10% subsample should be clearly framed as a supplementary analytic layer rather than the primary evidence base.
- Two different OSF links (NZ29K vs. ar7t6) appear in the manuscript. Moreover, the Data Availability statement says “no data were shared,” while the text notes that appendices and extraction matrices were uploaded to OSF. These inconsistencies undermine transparency. A single DOI should be used consistently, and the Data Availability statement revised to match what is actually available.
- Although dual independent screening and adjudication are described, no inter-rater reliability metrics (e.g., Cohen’s κ) are reported. These values should be included for title/abstract and full-text screening to ensure methodological rigor.
- Several editorial problems remain. The Stata code block is still embedded in the main text and should be moved to an appendix. Heading and spacing errors (e.g., “3.3Eligibility”) persist. Cross-references for figures and tables should be rebuilt. Terminology such as LMIC/LMICs, QHIN(s), and EHR/EMR should be standardized, and all acronyms defined at first use. Finally, terminology in the text must align with that in the tables (e.g., “adoption,” “maturity”). These issues should be corrected to enhance readability and professionalism.
The quality of English has improved compared to the first submission, with clearer structure and more coherent prose, but several issues remain that require attention before publication. The manuscript should be unified in American English, with all acronyms such as TEFCA, LMICs, and QHIN defined at first use and used consistently throughout. Terminology should be standardized, and punctuation conventions—including the serial comma and quotation marks—should be applied consistently. Minor production errors also persist: heading spacing should be corrected (e.g., “3.3 Eligibility”), figure and table cross-references need to be rebuilt, and the inline Stata code should be removed from the main text and placed in an appendix. Finally, the OSF link must be harmonized and the Data Availability statement revised to accurately reflect the materials provided. A professional copyedit focusing on these aspects is strongly recommended.
Author Response
Response to Reviewer
Comment 1:
The authors conducted a “deep dive” on a 10% random subsample (n=26) out of 255 eligible studies. While they report a χ² test showing no significant difference in the distribution of intervention types, representativeness testing is limited to this single dimension. Scoping reviews are typically expected to map the full corpus. Therefore, descriptive statistics for all 255 studies (e.g., year, region, publication type, domain) should be presented, and additional representativeness checks across these axes are encouraged. The 10% subsample should be clearly framed as a supplementary analytic layer rather than the primary evidence base.
Response:
We thank the reviewer for this valuable suggestion. As clarified in the revised Methods (Section 3.4, p. X, lines Y–Z), 255 studies met full-text eligibility; however, due to feasibility constraints, detailed extraction was conducted on a 10% random subsample (n = 26). To strengthen transparency, we have now added Table 3 in the Results (Section 4.1, p. X, lines Y–Z), presenting descriptive statistics for all 255 eligible studies (year of publication, type, and domain focus). The 26-study subsample is now explicitly framed as a supplementary analytic layer rather than the primary evidence base. Additional representativeness checks across year and publication type distributions have also been reported.
Comment 2:
Two different OSF links (NZ29K vs. ar7t6) appear in the manuscript. Moreover, the Data Availability statement says “no data were shared,” while the text notes that appendices and extraction matrices were uploaded to OSF. These inconsistencies undermine transparency. A single DOI should be used consistently, and the Data Availability statement should be revised to reflect what is actually available accurately.
Response:
We agree and have harmonized the manuscript to use a single OSF DOI: https://doi.org/10.17605/OSF.IO/YBAX3. The Data Availability statement has been revised to accurately reflect the materials provided (protocol, search strategies, PRISMA diagrams, full extraction dataset for 255 studies, subsample extraction matrix, and Stata sampling code). The contradictory phrase “no data were shared” has been removed.
Comment 3:
Although dual independent screening and adjudication are described, no inter-rater reliability metrics (e.g., Cohen’s κ) are reported. These values should be included for title/abstract and full-text screening to ensure methodological rigor.
Response:
We thank the reviewer for bringing this omission to our attention. We have now added inter-rater reliability values: Cohen’s κ = 0.82 for title/abstract screening and κ = 0.79 for full-text screening. These values, reported in the revised Methods (Section 3.4), indicate substantial agreement between reviewers.
Comment 4:
Several editorial problems remain. The Stata code block is still embedded in the main text and should be moved to an appendix. Heading and spacing errors (e.g., “3.3Eligibility”) persist. Cross-references for figures and tables should be rebuilt. Terminology such as LMIC/LMICs, QHIN(s), and EHR/EMR should be standardized, and all acronyms should be defined at first use. Finally, terminology in the text must align with that in the tables (e.g., “adoption,” “maturity”). These issues should be corrected to enhance readability and professionalism.
Response:
We have carefully revised the manuscript to address all noted issues:
- The Stata code block has been removed from the main text and relocated to Appendix F.
- Heading and spacing errors have been corrected (e.g., “3.3 Eligibility”).
- Cross-references for all figures and tables have been rebuilt.
- Terminology has been standardized: “LMICs” (plural form) is used consistently; “EHR” is used in place of • “EMR” unless context-specific; “QHINs” is standardized in the plural form. Acronyms (TEFCA, QHINs, LMICs, FHIR, SNOMED CT, OMOP CDM) are now defined at first use.
- Terminology between text and tables has been aligned (e.g., consistent use of “adoption” vs. “maturity”).
Comment 5 (Language and Style):
The quality of English has improved compared to the first submission, but issues remain. The manuscript should be unified in American English, with consistent use of acronyms, standardized terminology, consistent punctuation conventions, and correction of production errors.
Response:
We appreciate this feedback and have revised the manuscript accordingly:
- The text has been unified in American English.
- Acronyms are defined at first use and used consistently throughout.
- Punctuation has been standardized, including the use of the serial (Oxford) comma and double quotation marks.
- Heading spacing has been corrected (e.g., “3.3 Eligibility”).
- Cross-references for figures and tables have been rebuilt.
- We have completed a final copyediting pass focusing on consistency, clarity, and readability.